# A Missense Variant Affecting the C-Terminal Tail of UNC93B1 in Dogs with Exfoliative Cutaneous Lupus Erythematosus (ECLE)

**DOI:** 10.3390/genes11020159

**Published:** 2020-02-03

**Authors:** Tosso Leeb, Fabienne Leuthard, Vidhya Jagannathan, Sarah Kiener, Anna Letko, Petra Roosje, Monika M. Welle, Katherine L. Gailbreath, Andrea Cannon, Monika Linek, Frane Banovic, Thierry Olivry, Stephen D. White, Kevin Batcher, Danika Bannasch, Katie M. Minor, James R. Mickelson, Marjo K. Hytönen, Hannes Lohi, Elizabeth A. Mauldin, Margret L. Casal

**Affiliations:** 1Institute of Genetics, Vetsuisse Faculty, University of Bern, 3001 Bern, Switzerland; fabileuthard@gmail.com (F.L.); vidhya.jagannathan@vetsuisse.unibe.ch (V.J.); sarah.kiener@vetsuisse.unibe.ch (S.K.); anna.letko@vetsuisse.unibe.ch (A.L.); 2Dermfocus, University of Bern, 3001 Bern, Switzerland; petra.roosje@vetsuisse.unibe.ch (P.R.); monika.welle@vetsuisse.unibe.ch (M.M.W.); 3Division of Clinical Dermatology, Department of Clinical Veterinary Medicine, Vetsuisse Faculty, University of Bern, 3001 Bern, Switzerland; 4Institute of Animal Pathology, Vetsuisse Faculty, University of Bern, 3001 Bern, Switzerland; 5ZNLabs Veterinary Diagnostics, Garden City, ID 83714, USA; katherine@znlabs.com; 6Westvet, Garden City, ID 83714, USA; cannonderm@sbcglobal.net; 7AniCura Tierärztliche Spezialisten, 22043 Hamburg, Germany; monikalinek@gmail.com; 8Department of Small Animal Medicine and Surgery, College of Veterinary Medicine, University of Georgia, Athens, GA 30602, USA; fbanovic@uga.edu; 9Department of Clinical Sciences, College of Veterinary Medicine, North Carolina State University, Raleigh, NC 27607, USA; tolivry@ncsu.edu; 10Department of Medicine and Epidemiology, School of Veterinary Medicine, University of California Davis, Davis, CA 95616, USA; sdwhite@ucdavis.edu; 11Department of Population Health and Reproduction, School of Veterinary Medicine, University of California, Davis, CA 95616, USA; klbatcher@ucdavis.edu (K.B.); dlbannasch@ucdavis.edu (D.B.); 12Department of Veterinary and Biomedical Sciences, University of Minnesota, Saint Paul, MN 55108, USA; minork@umn.edu (K.M.M.); micke001@umn.edu (J.R.M.); 13Department of Veterinary Biosciences, University of Helsinki, 00014 Helsinki, Finland; marjo.hytonen@helsinki.fi (M.K.H.); hannes.lohi@helsinki.fi (H.L.); 14Department of Medical and Clinical Genetics, University of Helsinki, 00014 Helsinki, Finland; 15Folkhälsan Research Center, 00290 Helsinki, Finland; 16School of Veterinary Medicine, University of Pennsylvania, Philadelphia, PA 19104, USA; emauldin@vet.upenn.edu (E.A.M.); casalml@vet.upenn.edu (M.L.C.)

**Keywords:** *Canis familiaris*, dermatology, immunology, animal model, skin, TLR7, toll-like receptor, syndecan binding protein, syntenin-1, systemic lupus erythematosus, SLE, CLE

## Abstract

Cutaneous lupus erythematosus (CLE) in humans encompasses multiple subtypes that exhibit a wide array of skin lesions and, in some cases, are associated with the development of systemic lupus erythematosus (SLE). We investigated dogs with exfoliative cutaneous lupus erythematosus (ECLE), a dog-specific form of chronic CLE that is inherited as a monogenic autosomal recessive trait. A genome-wide association study (GWAS) with 14 cases and 29 controls confirmed a previously published result that the causative variant maps to chromosome 18. Autozygosity mapping refined the ECLE locus to a 493 kb critical interval. Filtering of whole genome sequence data from two cases against 654 controls revealed a single private protein-changing variant in this critical interval, *UNC93B1*:c.1438C>A or p.Pro480Thr. The homozygous mutant genotype was exclusively observed in 23 ECLE affected German Shorthaired Pointers and an ECLE affected Vizsla, but absent from 845 controls. UNC93B1 is a transmembrane protein located in the endoplasmic reticulum and endolysosomes, which is required for correct trafficking of several Toll-like receptors (TLRs). The p.Pro480Thr variant is predicted to affect the C-terminal tail of the UNC93B1 that has recently been shown to restrict TLR7 mediated autoimmunity via an interaction with syndecan binding protein (SDCBP). The functional knowledge on UNC93B1 strongly suggests that p.Pro480Thr is causing ECLE in dogs. These dogs therefore represent an interesting spontaneous model for human lupus erythematosus. Our results warrant further investigations of whether genetic variants affecting the C-terminus of UNC93B1 might be involved in specific subsets of CLE or SLE cases in humans and other species.

## 1. Introduction

In humans, cutaneous lupus erythematosus (CLE) represents a group of lupus erythematosus (LE)-associated autoimmune skin diseases exhibiting a cell-rich interface dermatitis leading to erosions and ulcerations with subsequent scarring, disfiguration and decreased quality of life [1,2,3,4]. CLE can affect only the skin or be present as part of a diverse range of potentially life-threatening and debilitating symptoms in patients with systemic lupus erythematosus (SLE) [1,2,3,4].

The incidence of CLE has been reported at ~4 cases per 100,000 persons per year [5,6,7,8]; 10% to 30% of human patients with CLE exhibit a transition from cutaneous into SLE forms, suggesting shared pathways and genetic background relevant to both cutaneous and systemic manifestations [5,6,9].

It has been proposed that some CLE forms, similarly to SLE, have an underlying genetic predisposition that combines with environmental factors to elicit an abnormal immune response with a continuous activation of the innate immune system. Several genetic associations have been identified in human CLE, with the majority of them involving type I interferon pathways, cell death and clearance of cell debris, antigen presentation and immune cell regulation [10,11]. To date, a single monogenic form of CLE caused by heterozygous variants in the *TREX1* gene encoding the three prime repair exonuclease has been identified in human patients with familial chilblain lupus erythematosus [12]. The pathogenic *TREX1* variants lead to chronic hyperactivation of the type I interferon system via cytosolic DNA recognition pathways [11,13]. A rare monogenic form of SLE in humans is caused by variants in the *DNASE1* gene encoding deoxyribonuclease 1 [14]. Mice deficient for Dnase I also develop an SLE-like autoimmune disease [15].

Dogs may also suffer from various forms of CLE, some of which resemble or are identical to their human homologs [4]. The so-called exfoliative cutaneous lupus erythematosus (ECLE) is a dog-specific variant of chronic CLE that has a very strong hereditary component and appears to be inherited as a monogenic autosomal trait [16,17,18]. Despite its current designation, signs of ECLE are not restricted to the skin. In most patients, ECLE starts with characteristic skin lesions in juvenile or young adult dogs (Figure 1). In later stages, ECLE often additionally affects the joints with severe pain, but a progression to classic antinuclear antibody-positive SLE is usually not seen [4,16,17,18]. The treatment of ECLE-affected dogs with immunomodulatory drugs often is insufficient to achieve long-lasting control of the disease, leading to a guarded prognosis [18,19]. Dogs affected with ECLE often are euthanized due to the severity of their disease. ECLE has been observed in several closely related hunting dog breeds, German Shorthaired Pointers, Braques du Bourbonnais, and Vizslas.

A previously reported genome-wide association study (GWAS) mapped the causative genetic defect for ECLE to chromosome 18, but the causative variant has not yet been identified [20]. The best-associated marker was located at position 53,913,829 (CanFam 2) [20], which corresponds to 50,888,317 in the current CanFam 3.1 assembly.

In the present study, we performed a new GWAS followed by a whole genome sequencing approach with the goal to identify the causative genetic variant for ECLE in dogs.

## 2. Materials and Methods 

### 2.1. Ethics Statement

All the dogs in this study were privately owned and samples were collected with the consent of their owners. The collection of blood samples was approved by the “Cantonal Committee for Animal Experiments” (Canton of Bern; permit 75/16).

### 2.2. Animal Selection

This study included 877 dogs. They consisted of 552 German Shorthaired Pointers (26 ECLE cases/526 controls), 52 unaffected German Longhaired Pointers, 210 unaffected German Wirehaired Pointers, 7 unaffected Braques du Bourbonnais, and 56 Vizslas (1 ECLE case/55 controls). The 27 ECLE cases were diagnosed by licensed veterinarians. The 850 dogs classified as unaffected represented population controls without reports of severe immunological or skin-related health issues. Peripheral blood samples were collected in EDTA vacutainers and stored at −20°C. Additional details on samples are given in Appendix A.

### 2.3. DNA Extraction and SNV Genotyping

Genomic DNA was either available from a previous study [20], isolated from EDTA blood with the Maxwell RSC Whole Blood Kit using a Maxwell RSC instrument (Promega, Dübendorf, Switzerland), or from formalin-fixed paraffin-embedded (FFPE) tissue samples using the Maxwell RSC DNA FFPE kit according to the manufacturer’s instructions. DNA from 14 ECLE cases and 29 controls was genotyped on illumina_HD canine BeadChips containing 220,853 markers (Neogen, Lincoln, NE, USA). The raw SNV genotypes are available in Appendix A. We did not have complete pedigree information on all 43 dogs that were genotyped on the SNV arrays. Some of the dogs were closely related, including, for example, 5 cases that were full siblings. Appendix A lists the pairwise IBD between all dogs and gives an objective measure of the relatedness between the genotyped dogs. A multiple dimension scaling (MDS) plot is shown in Appendix A.

The previously published GWAS [20] had been done with Affymetrix v2 127 k SNV genotyping arrays. A total of 6 cases and 2 controls were shared between the two analyses. The other 35 samples herein were from dogs different from those of the previous study.

For some dogs from the previous study [20] only very little DNA was left. The remaining DNA of 8 German Shorthaired Pointers was used up for SNV genotyping on the illumina_HD canine BeadChips. In these dogs, no specific targeted genotyping could be performed (see Section 2.8 below).

### 2.4. GWAS and Autozygosity Mapping

We used PLINK v.1.9 for basic file manipulation of the SNV genotypes [21]. We removed markers and individuals with less than 90% call rates. We further removed markers with minor allele frequency of less than 10% and markers deviating from the Hardy–Weinberg equilibrium in controls with a p-value of less than 10^−5^. An allelic GWAS was then performed with the GEMMA 0.98 software using a linear mixed model including an estimated kinship matrix as covariable to correct for the genomic inflation [22]. Manhattan and QQ plots of the corrected p-values were generated in R [23].

For autozygosity mapping, the genotype data of 14 ECLE cases were used. A tped-file containing the markers on chromosome 18 was visually inspected in an Excel spreadsheet to find a homozygous shared haplotype in the cases (Appendix A).

### 2.5. Whole Genome Sequencing of Two Affected German Shorthaired Pointers

Illumina TruSeq PCR-free DNA libraries with ~450 bp insert size of two affected German Shorthaired Pointers without known relationships were prepared. We collected 277 and 160 million 2 × 150 bp paired-end reads on a NovaSeq 6000 instrument corresponding to 29.3× and 17.9× coverage, respectively. Mapping and alignment were performed as described previously [24]. The sequence data were deposited under study accession PRJEB16012 and sample accessions SAMEA5657398 and SAMEA6249504 at the European Nucleotide Archive.

### 2.6. Variant Calling

Variant calling was performed using GATK HaplotypeCaller [25] in gVCF mode as described [24]. To predict the functional effects of the called variants, SnpEff [26] software together with NCBI annotation release 105 for the CanFam 3.1 genome reference assembly was used. For variant filtering we used 654 control genomes, which were either publicly available [27,28] or produced during other projects of our group [24] (Appendix A).

### 2.7. Gene Analysis

We used the CanFam 3.1 dog reference genome assembly and NCBI annotation release 105. Numbering within the canine *UNC93B1* gene corresponds to the NCBI RefSeq accession numbers XM_540813.6 (mRNA) and XP_540813.3 (protein).

### 2.8. Sanger Sequencing

The *UNC93B1*:c.1438C>A variant was genotyped by direct Sanger sequencing of PCR amplicons. On high-quality genomic DNA samples, a 399 bp PCR product was amplified from genomic DNA using AmpliTaqGold360Mastermix (Thermo Fisher Scientific, Waltham, MA, USA) together with primers 5‘-ATC CGT GTC TGT GCC CTC A-3‘ (Primer F) and 5’-CGA CCT GAG ACG CGG TAA A-3’ (Primer R). For FFPE-derived DNA samples, a smaller amplicon of 124 bp was amplified with the primers 5’-CCT CGT ACC TGT GGA TGG AG-3’ (Primer F2) and 5’-CTC TCG TCG GAG TTG TCC TC-3’ (Primer R2). After treatment with exonuclease I and alkaline phosphatase, amplicons were sequenced on an ABI 3730 DNA Analyzer (Thermo Fisher Scientific). Sanger sequences were analyzed using the Sequencher 5.1 software (GeneCodes, Ann Arbor, MI, USA).

## 3. Results

### 3.1. Mapping of the ECLE Locus

We performed a GWAS with genotypes from 43 German Shorthaired Pointers. After quality control, the pruned dataset consisted of 14 ECLE cases, 29 controls and 116,891 markers. We obtained a single association signal with 35 markers exceeding a suggestive significance threshold of *p* = 5 × 10^−5^ after adjustment for genomic inflation. All associated markers were located on chromosome 18 within an interval spanning from 49.0 Mb–53.9 Mb. The top-associated marker at Chr18:49,835,345 had a *p*-value of 1.5 × 10^−6^ (Figure 2).

To narrow down the identified region, we visually inspected the genotypes of the cases and performed an autozygosity mapping. We searched for homozygous regions with allele sharing and found one region of ~493 kb which was shared between all 14 cases. The critical interval for the causative ECLE variant corresponded to the interval between the first flanking heterozygous markers on either side of the homozygous segment or Chr18:49,545,431-50,038,225 (CanFam 3.1 assembly).

### 3.2. Identification of a Candidate Causative Variant

We sequenced the genome of two affected dogs at 29.3 × and 17.9 × coverage and called SNVs and small indel variants with respect to the reference genome. We then compared these variants to whole genome sequence data of 8 wolves and 646 control dogs from genetically diverse breeds. This analysis identified two private homozygous variants in the critical interval in the affected dogs (Table 1). A visual inspection of the short read alignments ruled out any additional structural variants affecting protein-coding sequences in the critical interval in the two sequenced cases.

One of the two private variants in the critical interval, Chr18:49,733,311C>T, was located in an intron of the *CHKA* gene and thus not investigated further. The other variant, Chr18:49,834,825C>A was a missense variant in the last exon of the *UNC93B1* gene. The formal designation of this variant is XM_540813.6:c.1438C>A or XP_540813.3:p.(Pro480Thr). It is predicted to change a highly conserved amino acid in the C-terminal tail of the UNC93B1 protein. We confirmed the variant by Sanger sequencing (Figure 3).

### 3.3. Genotype Phenotype Association of the UNC93B1:p.Pro480Thr Variant

We genotyped 544 German Shorthaired Pointers for the p.Pro480Thr variant and found a near perfect association of the genotypes at this variant with ECLE (*p_Fisher_* = 1.2 × 10^−39^). None of the 520 genotyped controls were homozygous for the mutant A/A genotype. However, one of the 24 genotyped cases was not homozygous A/A. We speculate that this single discordant dog is likely due to a phenotyping error as it had an atypically late age of onset and was not clinically confirmed as having ECLE by a board certified veterinary dermatologist (Appendix A). The analysis of additional animals from related hunting dog breeds revealed the presence of the mutant allele in German Longhaired Pointers and Vizslas. A single ECLE-affected Vizsla also had the homozygous mutant A/A genotype (Table 2).

## 4. Discussion

In this study, we identified UNC93B1:pPro480Thr as a candidate causative variant for ECLE in dogs. We performed a new GWAS and obtained the strongest association signal on the same chromosome, but approximately 1 Mb more proximal than the location in the previously reported GWAS [20]. Given that linkage disequilibrium within breeds can span several Mb, we consider our new result a confirmation and refinement of the previously reported association. Compared to the previous study [20], we detected a different ~500 kb homozygous haplotype block harboring the 7 top markers of our GWAS that was shared among all 14 investigated ECLE cases.

Whole genome sequencing data of two cases and 654 controls revealed a single private protein changing variant in the critical interval, UNC93B1:pPro480Thr. All but one of the designated ECLE cases were homozygous for the mutant allele with the single discordant dog believed to represent a phenotype mismatch. Conversely, the mutant allele was not found in a homozygous state in more than 1000 control dogs.

The mutant allele was also detected in heterozygous status in controls of two related breeds, German Longhaired Pointers and Vizslas. These breeds share a common ancestry with German Shorthaired Pointers. This provides indirect support for the previous observation that ECLE also can affect dogs from breeds related to the German Shorthaired Pointer. The hypothesis of a common genetic defect in these breeds was confirmed by our finding of an ECLE affected Vizsla that was also homozygous mutant at the *UNC93B1* variant.

The *UNC93B1* gene encodes a protein named “unc-93 homolog B1, TLR signaling regulator”. The human UNC93B1 consists of 597 amino acids and is a 12 transmembrane domain containing protein located in endosomal membranes [31]. It acts as a trafficking chaperone of the intracellular nucleic acid-sensing Toll-like receptors (TLRs) 3, 7 and 9 [32,33,34,35]. These TLRs are essential components of the innate immune system and activated when pathogen derived nucleic acids appear in endolysosomes. *UNC93B1* mediates the correct trafficking and localization of these TLRs to endolysosomes [32]. Complete loss-of-function of UNC93B1 results in a severe immune deficiency in human patients [36] and the *3d* mouse mutant [37].

Recently, the molecular mechanisms of the interaction of UNC93B1 with TLRs were studied in great detail. A 33 amino acid sequence motif in the cytoplasmic C-terminal domain of UNC93B1 binds to syndecan binding protein (SDCBP), also called syntenin-1. SDCBP interacts with both UNC93B1 and TLR7 [30]. This interaction dampens TLR7 signaling and prevents autoimmune activation of TLR7 by endogenous nucleic acids [30,35]. Gene-edited mice expressing a mutant Unc93B1 in which three critical amino acids of this C-terminal domain were altered (530-PKP/AAA-532) developed hallmarks of systemic inflammation and autoimmunity [30], similar to what has been observed in Tlr7 overexpressing mice [38,39,40]. In summary, the available literature suggests that complete loss of function of UNC93B1 leads to an immune deficiency, while UNC93B1 variants that only affect the C-terminal tail containing the SDCBP binding domain lead to upregulation of TLR7 signaling with subsequent development of systemic autoimmune disease.

The detailed functional knowledge on the role of the C-terminal tail of UNC93B1 for the regulation of TLR7 signaling strongly suggests that ECLE in dogs is due to dysregulated TLR7 signaling caused by the canine UNC93B1:p.Pro480Thr variant.

To the best of our knowledge, ECLE affected dogs represent the first spontaneous *UNC93B1* mutant that develops an autoimmune disease of the lupus group. Therefore, these dogs represent an interesting model for human CLE and/or SLE. As already suggested by [30], it seems possible that lupus erythematosus or other related autoimmune diseases in some human patients might be due to comparable genetic variants in *UNC93B1*.

## 5. Conclusions

We identified the spontaneously arisen UNC93B1:p.Pro480Thr variant as likely causative for ECLE in dogs. Knowledge of this variant will facilitate genetic testing of dogs to prevent the non-intentional breeding of ECLE-affected dogs. This unique canine form of CLE in dogs represents an interesting model for lupus erythematosus and potentially other autoimmune diseases in humans.

## Figures and Tables

**Figure 1 genes-11-00159-f001:**
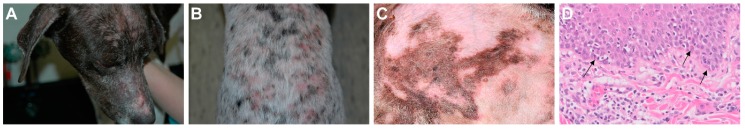
Exfoliative Cutaneous Lupus Erythematosus (ECLE) phenotype. (**A**) Scarring alopecia, generalized hair loss and adherent crusts on the face of a 2-year-old male dog. (**B**) Erythematous lesions on the back of a 1.5-year old male dog. (**C**) Close up of patchy lesions on the abdomen. (**D**) Haired skin from an ECLE affected dog with typical histological changes that include a cell-rich interface inflammation with frequent basal keratinocyte apoptosis (arrows). Hematoxylin and eosin stain.

**Figure 2 genes-11-00159-f002:**
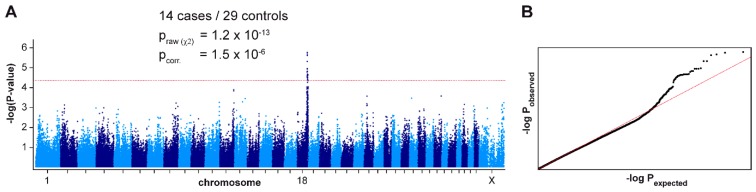
Mapping of the ECLE locus by genome-wide association. (**A**) Manhattan plot illustrating a single signal on chromosome 18. The dashed red line indicates the threshold for suggestive significance at *p* = 5 × 10^−5^ according to [29]. The best associated marker did not reach the stringent Bonferroni significance threshold (*p*_Bonf._ = 4.3 × 10^−7^) due to several close relationships and extreme genomic inflation in the dataset. The genomic inflation factor was 1.90 before and 0.99 after the correction. (**B**) The quantile–quantile (QQ) plot shows the observed versus expected –log(*p*) values. The straight red line in the QQ plot indicates the distribution of p-values under the null hypothesis. The deviation of *p*-values at the right side indicates that these markers are stronger associated with the trait than would be expected by chance. This supports the biological significance of the association.

**Figure 3 genes-11-00159-f003:**
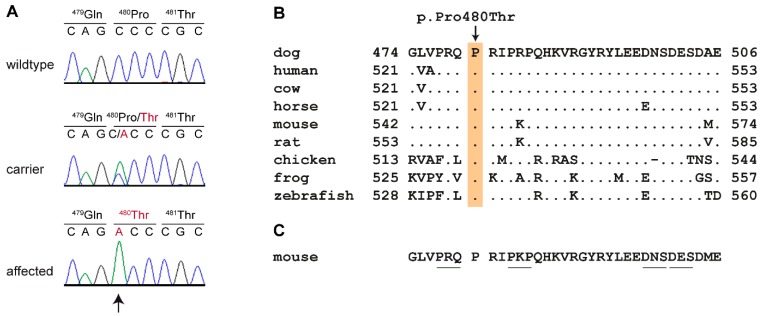
Details of the *UNC93B1* missense variant. (**A**) Representative Sanger electropherograms from dogs with the three different genotypes at c.1438C>A are shown. The amino acid translation is indicated. (**B**) Evolutionary conservation of the SDCBP binding domain [30]. The proline at position 480 of the canine UNC93B1 protein is strictly conserved across all vertebrates. The sequences were derived from the following database accessions: dog XP_540813.3, human NP_112192.2, cow XP_540813.3, horse XP_023510352.1, mouse NP_062322.2, rat NP_001101983.1, chicken XP_004941322.1, frog NP_001093723.1, zebrafish XP_0026660582.1. (**C**) Scanning-alanine mutagenesis in mouse macrophages identified four mutants that disrupt SDCBP binding and lead to upregulated TLR7 signaling [30]. The altered residues in these mutants are underlined.

**Table 1 genes-11-00159-t001:** Variants detected by whole genome re-sequencing of two ECLE-affected dogs.

Filtering Step	Variants
Shared homozygous variants in whole genome	1,420,602
Private homozygous variants (absent from 654 control genomes) in whole genome	25
Shared homozygous variants in 493 kb critical interval	851
Private variants (absent from 654 control genomes) in critical interval	2
Protein changing private variants in critical interval	1

**Table 2 genes-11-00159-t002:** Association of the genotypes at *UNC93B1*:c.1438C>A with ECLE.

ECLE Phenotype	Breed	C/C	C/A	A/A
Affected	German Shorthaired Pointer (*n* = 24)	1	–	23
Control	German Shorthaired Pointer (*n* = 520)	457	63	–
Control	German Longhaired Pointer (*n* = 52)	50	2	–
Control	German Wirehaired Pointer (*n* = 210)	210	–	–
Control	Braque du Bourbonnais (*n* = 7)	7	–	–
Affected	Vizsla (*n* = 1)			1
Control	Vizsla (*n* = 56)	51	5	–

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
