# Peer review of "A Missense Variant Affecting the C-Terminal Tail of UNC93B1 in Dogs with Exfoliative Cutaneous Lupus Erythematosus (ECLE)"

_genes, 2020, doi:10.3390/genes11020159_

Round 1

Reviewer 1 Report

The manuscript by Leeb et al. describes use of SNP genotyping and whole-genome sequencing to identify a likely variant causative of exfoliative cutaneous lupus erythematosus in dogs.   Although functional evidence in dogs was not supplied, computational prediction of function, conservation of the residue across species, and implication of the gene in immune dysfunction provides a strong case for its implication.  The manuscript is written concisely and needs only minor improvement as suggested below.

The manuscript would benefit from a more detailed explanation of phenotyping used for these dogs.  Was the condition grossly diagnosed or was histology performed?  More detailed description would not only help support the conclusions but also could help human practitioners interested in comparative studies.  Could the method of phenotyping used explain the discordance of one dog’s genotype with phenotype?

The authors note “extreme genomic inflation,” which is troublesome.  Were there known relationships among the cases?   I do not necessarily think the result would change but this should be addressed further.  If phenotyped full-sibs to cases were included that should be noted and would provide additional support to the result.  The lambda value observed needs to be reported.  Similarly, were the two dogs sequenced related? 

It is not necessary to show the entirety of Chr 18 in Table S2.  Additional color-coding of variants would help the readers to visualize haplotype patterns.  At position 49,781,939, the haplotype is noted D – what does this indicate?

Author Response

(1)

The manuscript would benefit from a more detailed explanation of phenotyping used for these dogs.  Was the condition grossly diagnosed or was histology performed?  More detailed description would not only help support the conclusions but also could help human practitioners interested in comparative studies.  Could the method of phenotyping used explain the discordance of one dog’s genotype with phenotype?

Response: We thank the reviewer for this valuable comment. We now compiled the reliability of the diagnoses for each of the 27 cases in Table S1. It is indeed striking that the discordant case was not clinically confirmed by a board certified veterinary dermatologist. The skin biopsy of this dog was evaluated in a regular commercial diagnostic lab and not at one of the involved academic centers. Finally, the age of onset of the clinical signs in this dog occurred later than in all other cases of our study. This supports our hypothesis that this dog suffered from a genetically distinct disease and was incorrectly classified as an ECLE case.

(2)

The authors note “extreme genomic inflation,” which is troublesome.  Were there known relationships among the cases?   I do not necessarily think the result would change but this should be addressed further.  If phenotyped full-sibs to cases were included that should be noted and would provide additional support to the result.  The lambda value observed needs to be reported.  Similarly, were the two dogs sequenced related?

Response: Unfortunately, we did not have full pedigree information on all dogs in the study and were thus not able to reconstruct a full pedigree of all dogs in the study. There were a few known relationships including one family consisting of 5 affected and 2 unaffected full-siblings.

We estimated the pairwise IBD of the 43 dogs genotyped on SNV arrays and provide this information as Table S2. We also prepared an MDS plot and provide this as Figure S1. We added the genomic inflation factor before and after correction to the legend of Figure 2. The two sequenced dogs were not closely related and we added this information to the methods section.

(3)

It is not necessary to show the entirety of Chr 18 in Table S2.  Additional color-coding of variants would help the readers to visualize haplotype patterns.  At position 49,781,939, the haplotype is noted D – what does this indicate?

Response: In the revised version, this has become Table S3. We removed ~80% of the markers and give now the genotypes starting from 48 Mb until the end of the chromosome. We added the explanation for the allele abbreviations (D indicates a “deletion” at an indel variant). We chose not to perform additional coloring of haplotypes as this table gives unphased genotypes and the haplotype pattern in heterozygous regions may be confusing to readers who are not familiar with such tped-files. As Table S3 is an Excel-file, any interested reader may easily add custom coloring of specific alleles, if desired.

Reviewer 2 Report

The manuscript presents evidence for the association of exfoliative cutaneous lupus erythematosus (ECLE) in German Shorthair Pointer dogs. The number of samples is low (only 2 affected dogs were genome sequenced); however, the associated genotyping and the biological information are convincing that a good candidate causative mutation has been identified. The manuscript is very well written with a logical presentation.

1) The number of cases is small (n=26) but this is likely unavoidable with relatively rare conditions within a dog breed. However, it is not clear why SNP genotyping was undertaken in only 14 cases and 29 controls – an explanation for this choice of limited samples being genotyped should be included.

2) Were the cases and controls related? The analysis has used an appropriate estimated kinship matrix but some further information on the degree of relatedness or inbreeding would be useful.

3) The finding that this is a causative mutation for ECLE in dogs is somewhat overstated as testing in breeds other than German Shorthaired Pointers has not been done (i.e. cases in other breeds). Hence the statement on L210-211 and the abstract L46 should be changed to say it is a candidate in German Shorthaired Pointers rather than dogs. Further, a caveat that testing in other breeds is needed to confirm association with ECLE in those breeds; the conclusion currently suggests that genetic testing in dogs is warranted but this is likely premature in most breeds.

4) For the dog with ECLE but with a CC genotype, the authors state it is ‘believed to represent a phenotype mismatch’. Can it be clarified if there are phenotype differences to support this or if this is a suggested explanation by the authors to account for an unexpected genotype. (L205, L219)

L123 spelling error: Hardy

Author Response

(1)

The number of cases is small (n=26) but this is likely unavoidable with relatively rare conditions within a dog breed. However, it is not clear why SNP genotyping was undertaken in only 14 cases and 29 controls – an explanation for this choice of limited samples being genotyped should be included.

Response: Two considerations limited our selection of samples for the SNV genotyping: (1) Some dogs could not be included due to degradation and/or limited quantity of leftover DNA. (2) We tried to minimize the overlap between dogs used in the previously published GWAS (Wang et al. 2010) and the present study. As we were not able to avoid a small overlap in the sample cohorts, we did not explicitly state these arguments, but this is at least implied with our statements in lines 122-127.

(2)

Were the cases and controls related? The analysis has used an appropriate estimated kinship matrix but some further information on the degree of relatedness or inbreeding would be useful.

Response: Please see our response to the very similar comment no. 2 from reviewer 1. We added Figure S1 showing an MDS plot, Table S2 showing the pairwise IBD estimates for all dogs and we now explicitly state the genomic inflation factors before and after correction for the relatedness in our samples.

(3)

The finding that this is a causative mutation for ECLE in dogs is somewhat overstated as testing in breeds other than German Shorthaired Pointers has not been done (i.e. cases in other breeds). Hence the statement on L210-211 and the abstract L46 should be changed to say it is a candidate in German Shorthaired Pointers rather than dogs. Further, a caveat that testing in other breeds is needed to confirm association with ECLE in those breeds; the conclusion currently suggests that genetic testing in dogs is warranted but this is likely premature in most breeds.

Response: In the meantime, we obtained a sample from an additional ECLE-affected Vizsla. This dog was also homozygous for the mutant allele at the UNC93B1:c.1438C>A variant. We also tested more control dogs from the other hunting dog breeds. We think that the additional experimental data now sufficiently back our claims in the manuscript.

(4)

For the dog with ECLE but with a CC genotype, the authors state it is ‘believed to represent a phenotype mismatch’. Can it be clarified if there are phenotype differences to support this or if this is a suggested explanation by the authors to account for an unexpected genotype. (L205, L219)

Response: Please see our response to the very similar comment no. 1 from reviewer 1. The phenotype of the discordant dog is indeed slightly different from the other cases (later age of onset) and the reliability of diagnostic evidence in this dog is lower than for most of the other cases (Table S1).

(5)

L123 spelling error: Hardy

Response: We corrected the spelling error.